

# Simulating wind farm flows at hub height with 2D Reynolds-averaged Navier-Stokes simulations

Mads Baungaard[1], Takafumi Nishino[1], and Maarten Paul van der Laan[2]

[1]University of Oxford, Department of Engineering Science, Parks road, OX1 3PJ, United Kingdom
[2]Technical University of Denmark, DTU Wind and Energy Systems, Risø Campus, Frederiksborgvej 399, 4000 Roskilde, Denmark

**Correspondence:** Mads Baungaard (mads.baungaard@eng.ox.ac.uk)

**Abstract.** Wind turbines in an offshore wind farm typically have the same hub height, and in this case, the power of a wind farm could be predicted if the flow field in the horizontal 2D plane at the hub height is predicted accurately. Nevertheless, Reynolds-averaged Navier-Stokes (RANS) simulations of wind farm flows are predominantly made in full 3D domains, which are naturally more computationally expensive than 2D simulations. In this work, a systematic comparison is made between 2D and 3D RANS simulations of various wind farm configurations to assess the differences in computational cost and accuracy. For our numerical setup and the cases considered, which include layouts with up to 144 turbines, it is found that the 2D simulations are at least two orders of magnitude computationally cheaper than their corresponding 3D simulations, while the predicted farm power is within $-30\%$ to $15\%$ for all cases. Only minor, but necessary, modifications have been made to the governing 2D RANS equations to avoid unphysical decay of turbulence, allowing for a simple direct comparison between the 2D to 3D simulations. Given the low computational cost and already sensible performance of the only slightly modified 2D RANS simulations demonstrated in this work, it appears attractive to further investigate this methodology and possibly introduce additional 2D modifications to improve the accuracy in future work.



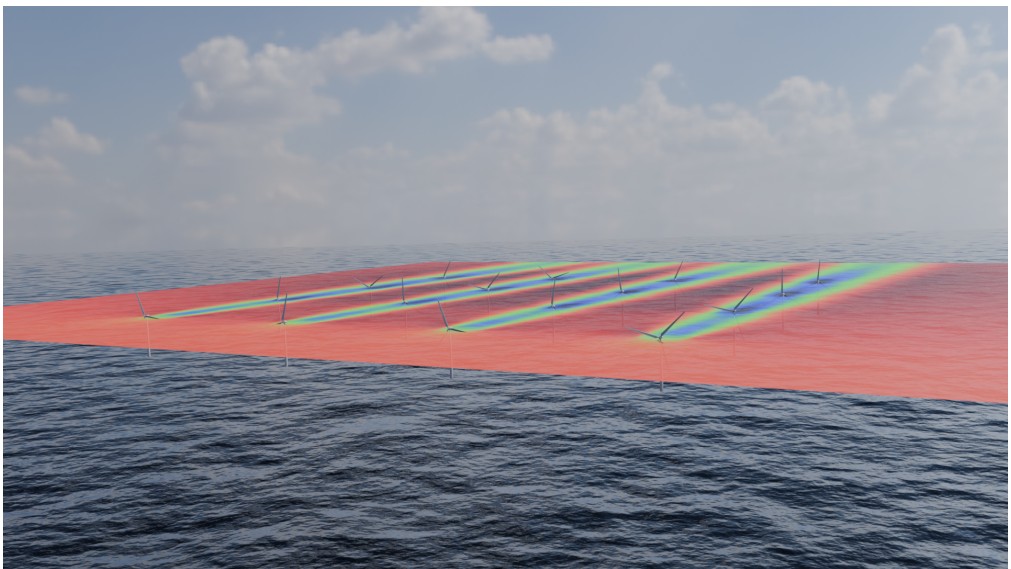

**Figure 0.** Graphical abstract.

# 1 Introduction

Turbine wakes may cause large power losses in wind farms, and computational tools can be used to understand and possibly
mitigate these. For wind farm flow simulations, the Reynolds-averaged Navier-Stokes (RANS) technique is generally consid-
ered a mid-fidelity tool, both in terms of computational cost and accuracy, somewhere between engineering models (EMs) and
large-eddy simulations (LESs). A 3D domain is typically used for RANS simulations (van der Laan et al., 2015c; Bleeg et al.,
2018; Zehtabiyan-Rezaie and Abkar, 2023) and LESs (Calaf et al., 2010), whereas a 2D domain at the wind turbine hub height
is more often used for the EMs (Pedersen and et al., 2023). While LES demands three-dimensionality to include the vortex
stretching processes necessary for properly resolving the inherently three-dimensional energy-carrying eddies (Piomelli, 2020),
there is no such requirement for RANS. Indeed, many of the early studies of RANS turbulence modeling were centered entirely
around 1D and 2D cases due to the limitation of computer power at the time (Jones and Launder, 1972). The consideration of
computing costs however still remains of importance today and although RANS simulations are significantly less expensive
than LES, they still require high-performance computing clusters or powerful workstations to simulate a large wind farm.

Inspired by the fast 2D EMs, we explore in this work the capability of 2D RANS simulations of wind farm flows. There
are essentially three types of 2D simulations possible for wind farm flows, (i) 2D flow in the $xy$-plane at hub-height, (ii)
2D flow in the $xz$-plane through the turbine center, and (iii) 2D axisymmetric flow in the $xr$-plane (where $x$, $y$, $z$ and $r$ are
the streamwise, lateral, vertical and radial coordinates, respectively). A fourth possibility is depth-averaged 2D simulations
(e.g. Letizia and Iungo, 2022). Several axisymmetric 2D simulations have been used to study wind turbine flows in the past
(Sørensen and Myken, 1992; Sørensen and Kock, 1995; Madsen, 1996; Mikkelsen et al., 2001), although mostly assuming
laminar and uniform inflow (often using no turbulence model) and focusing on a single turbine. Due to the axisymmetry





assumption it is not possible to include sheared inflow and furthermore it is not possible to investigate different wind turbine layouts, except a single row of turbines, with this type of 2D simulation. With option (ii) it is possible to include sheared wind profiles and the effect of the ground, but as with (iii), it is not possible to investigate different wind turbine layouts. In this

work, we will restrict ourselves to type (i), where it is possible to consider general turbine layouts, given that the turbines have equal hub heights and are situated in flat terrain. A summary of the three types is given in Table 1.

| 2D type | 3D flow | Vertical shear & ground | General turbine layout |
|---|---|---|---|
| (i) $xy$-plane | ✗ | ✗ | ✓ |
| (ii) $xz$-plane | ✗ | ✓ | ✗ |
| (iii) $xr$-plane | ✓ | ✗ | ✗ |

**Table 1.** Types of 2D simulations for wind farm flows.

The type (i) 2D simulations have not been utilized much for wind farm studies in the litterature, but one example is the study of Erek (2011), who implemented the $k$-$\varepsilon$ model in an in-house 2D RANS code and simulated the Wieringeermeer wind farm testcase. It was found that the wake recovers too quickly compared to the experimental results, which however is a known

defect of the standard $k$-$\varepsilon$ model (van der Laan et al., 2015d), but the overall wake trends were nevertheless captured. There is no mentioning of the problem of downstream decay of turbulence, which is an essential problem for type (i) 2D simulations employing two-equation turbulence models and will be discussed thoroughly in Sect. 2.3. This omission might be because relatively small domains were used, which partly hides the effect of unphysical decay of turbulence. Another example of type (i) 2D simulation is seen in the work of Boersma et al. (2018), who interestingly modified the 2D continuity equation and

found that this led to better agreement with their 3D LES data. The modification is based on an axisymmetry assumption (even though their 2D code is *not* axisymmetric), but no direct investigations about the validity of this strong assumption were reported. Both constant and time-varying inflow were used, and since they derived their model equations from the LES equations with an algebraic mixing-length sub-grid scale (SGS) closure, their model is perhaps more correctly labeled as a 2D LES model. In another related study (van den Broek et al., 2022), it was stated that 2D wakes in general are too wide and

induce a too large speed-up effect at the sides of the wake (aka. an increased bypass velocity caused by the local blockage effect (Nishino and Draper, 2015)) compared to 3D wakes and that these issues can be alleviated with the 2D continuity correction. In this work, we choose to not apply this correction or other a-priori 2D corrections and instead focus on the performance of the purely 2D (i.e. 2D mass conserved) RANS simulation.

The 2D RANS simulations can be classified as being somewhere between low-fidelity (EMs) and medium-fidelity (3D

RANS) models, since some flow information is obviously lost by removing the vertical dimension, for example the shear and ground effects (which however will be partly compensated for through turbulence source terms), but much of the flow physics are nevertheless still retained. An important point is that the pressure is still solved in 2D RANS, which sets it apart from EMs with empirical blockage models or parabolic RANS simulations. The overall objective of this paper is to assess whether 2D RANS simulations can give sensible results for wind farm flows and to determine how much computational resources can be





saved compared to full 3D simulations. The structure of the paper is as follows: in Sect. 2, the numerical setup is presented. Special care is given to describe the 2D modifications of the $k$-$\varepsilon$ model necessary to avoid unphyiscal decay of turbulence and the actuator disk (AD) concept in 2D. A brief description of the testcases are then given in Sect. 3, which include a single turbine case, a series of cases with increasingly large arrays of turbines, and some cases where the wind direction is not aligned with the rows of turbines. The results are shown and discussed in Sect. 4. Finally, the conclusions of the study are given in Sect. 5. A grid sensitivity study and a verification of the 2D AD implementation are included in the appendices.

## 2 Methodology

### 2.1 RANS equations

The steady-state RANS equations with the standard $k$-$\varepsilon$ turbulence model (Pope, 2000) are the basis for both 2D and 3D simulations,

$$\frac{\partial U_j}{\partial x_j} = 0, \tag{1}$$

$$U_j \frac{\partial U_i}{\partial x_j} = -\frac{1}{\rho}\frac{\partial P}{\partial x_i} - \frac{\partial \overline{u_i u_j}}{\partial x_j} + f_i, \tag{2}$$

$$U_j \frac{\partial k}{\partial x_j} = \underbrace{-\overline{u_i' u_j'}\frac{\partial U_i}{\partial x_j}}_{\mathcal{P}} - \varepsilon + \underbrace{\frac{\partial}{\partial x_j}\left(\frac{\nu_t}{\sigma_k}\frac{\partial k}{\partial x_j}\right)}_{\mathcal{D}_k} + S_k, \tag{3}$$

$$U_j \frac{\partial \varepsilon}{\partial x_j} = C_{\varepsilon 1}\frac{\varepsilon}{k}\mathcal{P} - C_{\varepsilon 2}\frac{\varepsilon^2}{k} + \frac{\partial}{\partial x_j}\left(\frac{\nu_t}{\sigma_\varepsilon}\frac{\partial \varepsilon}{\partial x_j}\right) + S_\varepsilon, \tag{4}$$

$$\overline{u_i' u_j'} = \underbrace{-C_\mu \frac{k^2}{\varepsilon}}_{\nu_t}\left(\frac{\partial U_i}{\partial x_j} + \frac{\partial U_j}{\partial x_i}\right) + \frac{2}{3}k\delta_{ij}, \tag{5}$$

which are the governing equations of the velocity, $U_i$, the pressure, $P$, the turbulent kinetic energy (TKE), $k$, and the dissipation of TKE, $\varepsilon$.

In the case of 2D simulations, which corresponds to setting $\partial/\partial z = 0$, we also introduce source terms to the $k$- and $\varepsilon$-equations, $S_k$ and $S_\varepsilon$, see Sect. 2.3. For the 3D simulations $S_k = S_\varepsilon = 0$. The turbulence coefficients used are described in Table 2 and are the values recommended for atmospheric flows by Želi et al. (2020). These were also used for wind turbine flow simulations in a previous study by Baungaard et al. (2022b).

| $C_\mu$ | $\kappa$ | $C_{\varepsilon 1}$ | $C_{\varepsilon 2}$ | $\sigma_k$ | $\sigma_\varepsilon$ |
|---------|----------|---------------------|---------------------|------------|----------------------|
| 0.087   | 0.38     | 1.44                | 1.82                | 1.0        | 1.3                  |

**Table 2.** Turbulence coefficients. Recommendation of Želi et al. (2020).





For the inflow, we use the atmospheric log-law

$$U(z) = \frac{u_*}{\kappa} \log\left(\frac{z}{z_0}\right), \tag{6}$$

$$k(z) = \frac{u_*^2}{\sqrt{C_\mu}}, \tag{7}$$

$$\varepsilon(z) = \frac{u_*^3}{\kappa z}. \tag{8}$$

Here $u_*$ is the friction velocity and $z_0$ is the aerodynamic roughness length, which can be set to obtain a desired velocity and turbulence intensity at hub height. The log-law equations describe the flow of a neutral atmospheric surface layer (ASL) and are thus a simplification of the real ABL. However, they have the advantages that they can be specified analytically at the inlet and are equilibrium solutions to the RANS equations. By "equilibrium", we here refer to that the insertion of Eq. 6-8 into the right-hand side (RHS) of Eq. 2-4 yields a zero on the left-hand side (LHS). In other words, the flow and turbulence will not

develop through an empty domain if the roughness is constant over the bottom surface. This is a highly desirable property for wind farm studies charecterized by large domains. However, this equilibrium property is only assured in 3D and some additional source terms will therefore need to be added in 2D, see Sect. 2.3.

   To model the wind turbines we employ actuator disks (ADs), which act on the RANS equations through a momentum source term, $f_i$, in Eq. 2. For simplicity, we neglect the influence of the ADs on the turbulence equations, although it can be large in

the turbine vicinity (Réthoré, 2009) and that a model of its contribution has recently been suggested by Zehtabiyan-Rezaie and Abkar (2023), which was shown to improve the prediction of wake TKE. The specifics of the AD implementation are given in Sect. 2.4.

## 2.2   Numerical setup

The open-source unstructured finite-volume CFD code OpenFOAM v2206 is used to solve the model equations described in

the previous section for both 2D and 3D cases and the numerical setup is similar thus enabling a direct comparison between the two. To run the cases, a simple Python framework is used, which is also open-source, see Code and data availability.

   Despite the unstructured grid capability of OpenFOAM, only recti-linear structured grids are used in this work, as are typical for RANS simulations of wind farm flows (van der Laan et al., 2015c; Baungaard et al., 2022b), and a sketch of a grid is shown in Fig. 1. Since several turbine layouts are simulated, each simulation has its own customized grid, which is designed such that

the farm is encompassed in a uniform inner domain. A resolution, similar to other earlier studies, of $\Delta x = \Delta y = D/8$ is used in this inner domain and was found to be a good compromise between accuracy and computational cost in a grid study, see Appendix A. The grid is stretched outwards from the inner domain to the outer domain edge, which is set $50D$ away, in order to (i) ensure that the artificial tunnel blockage effect is small, (ii) have space for any numerical flow development (although the log-law is theoretically an equilibrium solution it might not be exactly numerically) from the inlet to the inner domain, (iii) to

avoid farm-scale induction effects on the boundaries, (iv) have negligibly small gradients at the outlet, and (v) using stretching to reduce the otherwise excessively large number of mesh cells. A cell size of $5D$ is used in the outermost cells of the outer domain. In the vertical direction (only relevant for the 3D simulations), the grid is geometrically stretched, i.e. $\Delta z_{i+1} = r_s \Delta z_i$



(where $r_s$ is the grid streching ratio), using a first-cell height of $\Delta z_1 = D/40$, $N_z = 46$ number of cells and a domain height of $L_z = 25D$, which gives a last cell-height of $\Delta z_{N_z} = 2.5D$. All 3D simulations in this work have the same vertical grid levels.

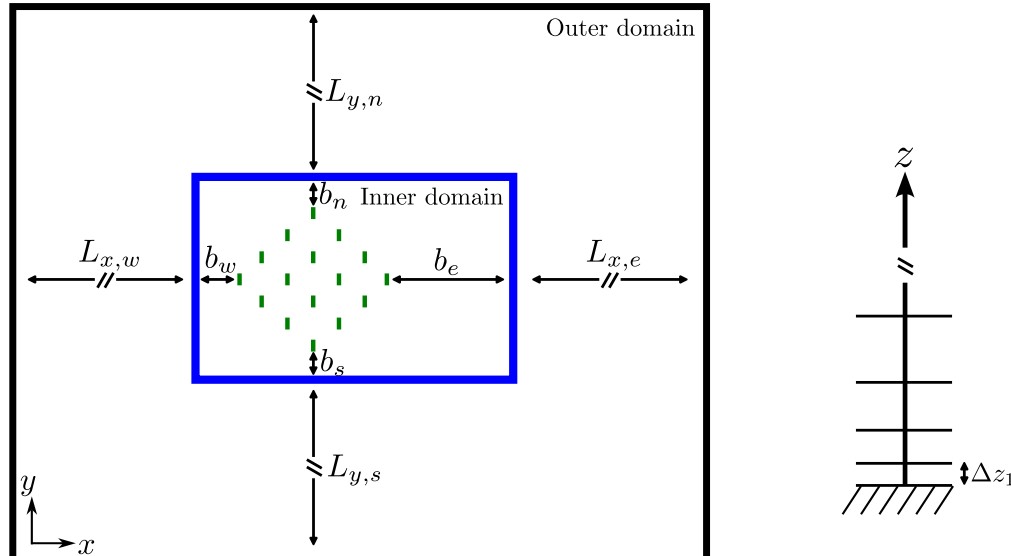

**Figure 1.** (left) Sketch of grid for a 45-degree rotated 4x4 farm. There is a buffer of $b_w = 3D$, $b_e = 9D$, $b_n = 2D$ and $b_s = 2D$ between the turbines and the inner domain edge. The outer domain edge is $L_{x,w} = L_{x,w} = L_{y,s} = L_{y,n} = 50D$ away from the inner domain. (right) Sketch of vertical grid levels.

A Dirichlet boundary condition (BC) is used at the inlet with the analytical log-law profiles for $U$, $k$ and $\varepsilon$, Eq. 6-8. Similarly, a Dirichlet BC is used at the top BC with the analytic log-law values evaluated at $z = L_z$. At the outlet a Neumann BC is set with a zero-gradient in the streamwise direction. On the two sides of the domain a symmetry BC is applied. Finally, a rough wall BC is set at the bottom boundary of the domain, which parametrizes the surface roughness using an aerodynamic roughness length $z_0$ (Hargreaves and Wright, 2007). For the flow initialization, we simply use a spatially uniform initialization of all

variables with the hub-height log-law values.

    The SIMPLE algorithm (simpleFoam in OpenFOAM) is used to solve the incompressible mass and momentum equations, Eq. 1-2, using Rhie-Chow interpolation to avoid checkerboard solutions that can else develop on collocated grids. To discretize the equations, we use a first-order accurate upwind scheme for the convective terms, while a second-order accurate Gaussian linear scheme (equivalent to the central difference scheme for structured grids) is used for the remainder of the derivatives in the

equations. This choice of discretization follows the turbineSiting tutorial of OpenFOAM v2206 and other earlier OpenFOAM wind turbine studies (Hornshøj-Møller et al., 2021; Zehtabiyan-Rezaie and Abkar, 2023). The resulting system of equations are solved sequentially (i.e. segregated) with a Gauss-Seidel solver for the $U_i$, $k$ and $\varepsilon$ equations, whereas a geometric algebraic multigrid (GAMG) solver is used for the pressure-correction equation. No grid sequencing technique is used to speed up convergence. A residual criterion of $10^{-3}$ is used for all equations, which was found to sufficiently converge the simulations,

see Appendix A.





## 2.3 Additional turbulence source terms for 2D simulations

As alluded in the earlier sections, additional source terms are needed in the turbulence equations for 2D RANS simulations, because the turbulence will otherwise decay. To understand this, consider an empty 2D domain, where we set $(U,V) = (U_\infty, 0)$ at the inlet. Since there are no turbines, symmetry BCs at the side of the domain and due to mass conservation, we must in fact have $(U,V) = (U_\infty, 0)$ in the whole domain. In this case, the turbulence equations without any source terms reduce to

$$U_\infty \frac{\partial k}{\partial x} = \underbrace{0}_{\mathcal{P}} - \varepsilon + \underbrace{\frac{\partial}{\partial x}\left(\frac{\nu_t}{\sigma_k}\frac{\partial k}{\partial x}\right)}_{\mathcal{D}_k}, \tag{9}$$

$$U_\infty \frac{\partial \varepsilon}{\partial x} = \underbrace{0}_{\mathcal{P}_\varepsilon} - C_{\varepsilon 2}\frac{\varepsilon^2}{k} + \underbrace{\frac{\partial}{\partial x}\left(\frac{\nu_t}{\sigma_\varepsilon}\frac{\partial \varepsilon}{\partial x}\right)}_{\mathcal{D}_\varepsilon}, \tag{10}$$

where it has been used that the flow is homogeneous in the $y$-direction, i.e. $\partial/\partial y = 0$ due to the symmetry BCs. The equations are similar to the classical case of turbulence decay often used to calibrate turbulence models (Pope, 2000), except that the diffusion terms here are retained. For a given inlet turbulence state, $(k,\varepsilon) = (k_\infty, \varepsilon_\infty)$, one can integrate the equations and an example is shown in Fig. 2. It can be seen that the TKE decays with more than 30% over just one kilometer, which is unphysical and problematic for wind farm simulations.

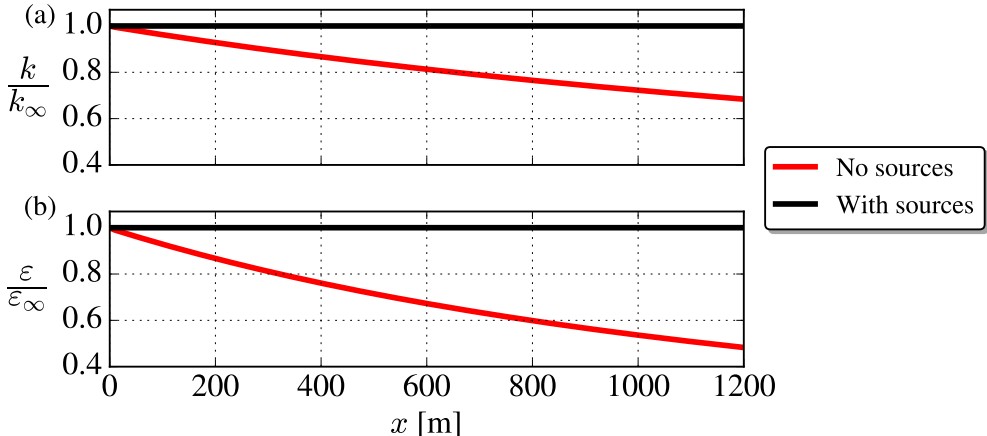

**Figure 2.** Turbulence development in an empty 2D domain. Inlet turbulence state is $K_\infty = 0.28 \text{ m}^2\text{s}^{-2}$ and $\varepsilon_\infty = 8.48\cdot 10^{-4} \text{ m}^2\text{s}^{-3}$, which corresponds to log-law flow with velocity $U = 8$ m/s and turbulence intensity $I = 5.4\%$ at $z = 70$ m.





In order to remedy this, one can modify the turbulence equations with some yet to be determined source terms

$$U_\infty \frac{\partial k}{\partial x} = \underbrace{0}_{\mathcal{P}} - \varepsilon + \underbrace{\frac{\partial}{\partial x}\left(\frac{\nu_t}{\sigma_k}\frac{\partial k}{\partial x}\right)}_{\mathcal{D}_k} + S_k, \tag{11}$$

$$U_\infty \frac{\partial \varepsilon}{\partial x} = \underbrace{0}_{\mathcal{P}_\varepsilon} - C_{\varepsilon 2}\frac{\varepsilon^2}{k} + \underbrace{\frac{\partial}{\partial x}\left(\frac{\nu_t}{\sigma_\varepsilon}\frac{\partial \varepsilon}{\partial x}\right)}_{\mathcal{D}_\varepsilon} + S_\varepsilon. \tag{12}$$

Inserting a constant solution ansatz, $k(x) = k_\infty$ and $\varepsilon(x) = \varepsilon_\infty$, into the above equations yields

$$S_k = \varepsilon_\infty, \tag{13}$$

$$S_\varepsilon = C_{\varepsilon 2}\frac{\varepsilon_\infty^2}{k_\infty}. \tag{14}$$

Solving the turbulence equations numerically with these source terms included indeed gives no turbulence decay, see also Fig.
2, and is a very simple modification. The idea of including turbulence source terms to promote flow equilibrium was also used
by van der Laan et al. (2017), although the source terms in their work are different and were designed to obtain equilibrium for
the the Monin-Obukhov similarity theory (MOST) profiles. In later work (Baungaard et al., 2022a) it was shown that in wind
turbine wakes these "equlibrium" source terms are negligible compared to other terms in the TKE budget, hence the source
terms do not have any significant influence on the wakes. A similar TKE and $\varepsilon$ budget analysis (not shown here) has been made
for the current study, which gives the same conclusion, namely that the source terms, $S_k$ and $S_\varepsilon$, have a negligible impact on
the wakes and only contribute to keeping the undisturbed flow in equilibrium. Similar source terms as Eq. 13-14 were also by
van der Laan et al. (2015a), although for the different purpose of enhancing numerical stability by artificially sustaining finite
turbulence quantities in the free atmosphere of their 3D wind farm simulations.

### 2.4 The Actuator Disk (AD) model

The wind turbines in this work are modeled with ADs (Calaf et al., 2010; van der Laan et al., 2015b), where the thrust and
power of each turbine are calculated as

$$T = \frac{1}{2}\rho A U_d^2 C_T', \tag{15}$$

$$P = \frac{1}{2}\rho A U_d^3 C_P'. \tag{16}$$

Here $\rho$ is the fluid density, $A$ is the area of the disk, $U_d$ is the disk-averaged velocity, $C_T'$ is the local thrust coefficient and $C_P'$
is the local power coefficient (note that these local coefficients are based on $U_d$). Importantly, these expressions do not depend
on the freestream velocity, which is generally not known for turbines within wind farms. The thrust and power coefficients in



the expressions can be related to the usual thrust coefficient $C_T$ (based on freestream velocity) through 1D momentum theory

$$a = \frac{1}{2}\left(1 - \sqrt{1 - C_T}\right),$$ (17)

$$C_P = 4a(1-a)^2,$$ (18)

$$C_T' = \frac{4}{a^{-1} - 1},$$ (19)

$$C_P' = C_T'.$$ (20)

In this study, we set the AD's to operate at $C_T = 0.75$, which is a typical value for below-rated wind speeds (van der Laan et al., 2022), and gives $a = 0.25$, $C_P = 0.56$ and $C_T' = C_P' = 1.33$. These coefficients are chosen a-priori and do not change during the simulation, which corresponds to assuming flat thrust and power curves. This is however reasonable, because they

indeed typically are flat below rated wind speed, except in the region between no wind and cut-in wind speed.

In the 3D simulations, each AD is defined in the code as a cylindrical volume with a thickness of two layers of cells, which encloses a number of grid cells where body forces, $f_i$ in Eq. 2, will be applied. This choice of thickness is made because choosing a thickness of only one cell leads to large spurious pressure fluctuations in the solution, unless a sophisticated algorithm is used to transform the momentum sources into pressure jumps (Réthoré and Sørensen, 2012; Troldborg et al., 2015).

The disk-averaged velocity, $U_d$, is obtained in each iteration through a straight-forward volume-average of the velocity in the cylindrical volume and the thrust and power can thereafter be calculated through Eq. 15-16. In each iteration, the thrust force is distributed uniformly in the AD volume with the force in a given cell being proportional to its volume divided by the total AD volume.

The AD concept is similar in 2D simulations, except that the actuator now has the shape of a line segment instead of a disc.

In the literature, the actuator is therefore also sometimes referred to as an actuator "strip" or "ribbon" (Réthoré and Sørensen, 2012). The only difference for 2D simulations is that the thrust is calculated using an area of $A = D$ (instead of $A = \pi(D/2)^2$ in 3D), which means that its unit is then force per unit depth. As a post-processing step, one can then estimate the equivalent 3D thrust and power as

$$T^{3D} = \frac{\pi D}{4} T^{2D},$$ (21)

$$P^{3D} = \frac{\pi D}{4} P^{2D}.$$ (22)

A 2D AD, in fact, represents an infinite ribbon in 3D, hence this estimate is in essence only a model, and it is intuitively clear that it will somehow deviate from the power predicted by a full 3D model. Specifically, since the flow in 2D can only divert around the sides of the turbine (and not above and below it as in 3D), it is expected that this will lead to a 2D blockage effect which increases the power of the turbine slightly more than the corresponding 3D blockage effect.

In order to verify that the 2D AD has been implemented correctly, we solve the 2D Euler equations for a turbine with a small $C_T$, which has an analytical solution (Madsen, 1988), see Appendix B.





# 3 Case descriptions

A series of case studies, summarized in Table 3, are conducted to assess the accuracy and computational cost of wind farm simulations with 2D and 3D RANS, respectively. All cases consider a Vestas V80 turbine with diameter $D = 80$ m, hub-height $z_h = 70$ m and thrust coefficient $C_T = 0.75$. Neutral log-law inflow with hub-height values $U_\infty = 8$ m/s and $I_\infty = 5.4\%$ is used for all cases, except the high turbulence intensity cases with $I_\infty = 10.0\%$. As described in Sect. 2.2, the same vertical discretization is used for all 3D cases, hence the grids only differ in the horizontal directions depending on the wind farm layout.

The first study is of a single turbine, where we in addition to the 2D and 3D OpenFOAM simulations also include results from an earlier 3D RANS simulation conducted with EllipSys3D (Baungaard et al., 2022b), which together with the study from Hornshøj-Møller et al. (2021) were the inspirations for this case setup. In the second case study, a total of 16 different turbine layouts are simulated, which are aligned rectangular layouts with $N_x = \{1, 4, 8, 12\}$ rows in the $x$-direction and $N_y = \{1, 4, 8, 12\}$ rows in the $y$-direction. The 16 layouts are also simulated with a lower array density and higher turbulence intensity, respectively. Finally, two of the layouts are simulated with a range of different wind directions (a wind direction of 0 degrees is in this study defined as being along the $x$-axis).

| Case study | Layouts | Inter-spacing $[D]$ | Wind direction $[°]$ | TI $[\%]$ |
|---|---|---|---|---|
| Single turbine | 1x1 | - | 0 | 5.4 |
| Turbine layouts | 1x1 - 12x12 | $\{4, 7\}$ | 0 | $\{5.4, 10.0\}$ |
| Wind directions | 8x8 & 8x4 | 4 | 0 - 90 | 5.4 |

**Table 3.** Overview of simulation cases.

For each study, the computational cost of the OpenFOAM 2D and 3D simulations will be reported in terms of CPU minutes (that is, number of cores × wall-clock time). In this regard, it should be emphasized that these absolute numbers are dependent on the computational hardware and the exact setup of the solver, but since the 2D and 3D simulations are set up similarly and run on the same hardware it can be expected that the observed speed-up is somewhat general. Several solver options (for example, the vertical discretization, residual criteria, cells per core or addition of grid sequencing) will likely change the computational speed-up, which could both become smaller or larger depending on these options.

Intentionally, reference experimental and LES data are not included in the following because the purpose is to assess how the 2D RANS results compare to corresponding 3D RANS results. By accuracy we therefore refer to how well 2D RANS matches with 3D RANS. Regarding the accuracy of 3D RANS itself, we instead refer to other studies (Prospathopoulos et al., 2011; van der Laan et al., 2015d; Bleeg et al., 2018; Baungaard et al., 2022b; Zehtabiyan-Rezaie and Abkar, 2023). The accuracy will mainly be reported in terms of the farm power, which can interestingly be under- or over-predicted depending on the farm layout, wind direction, array density and turbulence intensity as shall be documented in the following.





# 4   Results and discussion

## 4.1   Single turbine

The streamwise velocity fields at hub height of 2D and 3D simulations for the single turbine case are shown in Fig. 3. A 3D simulation with a fixed force AD (labelled as "3D-FF") is also included, which differs from the AD described in Sect. 2.4 in that the thrust is fixed a-priori. The data from an earlier study (Baungaard et al., 2022b) using the EllipSys3D code (labeled as "Reference 3D"), which also employed a FF-AD, is used to verify the current 3D OpenFOAM setup. It can be seen from the velocity contours that there is a qualitative good agreement between the 3D simulations in the far-wake, while a small

but noticeable difference is visible in the near-wake, especially for the 3D simulation based on the disk-averaged velocity AD (labeled as "3D"). This is more clearly illustrated with the wake profiles in Fig. 4 and the centerline velocity plot in Fig. 5. At $x/D = 1$, the difference in centerline velocity compared with the reference solution is 5.1% and 1.8% for the 3D and 3D-FF simulations, respectively, while at $x/D = 5$ the absolute difference is less than 0.6% for both. The main reason for the larger difference with the "3D" case is that the AD in the reference study was operated with a fixed thrust of $C_T = 0.77$

(which is therefore also used in the 3D-FF simulation), whereas the AD of the "3D" case uses Eq. 15 based on 1D momentum theory to calculate the thrust force, which for $C_T' = 1.33$ is supposed to give $C_T = 0.75$, but, in fact, gives a thrust coefficient of $C_T \approx 0.73$ for the current grid resolution, see Appendix A. The grid study in Appendix A also shows that the near-wake predictions can be improved with finer grid resolution, but at a significant increase in computational cost. As noted in Sect. 2.4, it is not possible to use a fixed force AD type in wind farm studies, which is why we opt for the 1D momentum based AD

in this study, although it gives a small error on $C_T$, $C_P$, and the flow field. Another error to note is the small velocity wiggle visible in Fig. 5 close to $x/D = 0$, which is caused by the force allocation method and decreases for finer grid resolution, see Appendix A. With these minor errors in mind and the else good comparison between the flow fields of the 3D simulations, we conclude that the 3D OpenFOAM setup is well-functioning.





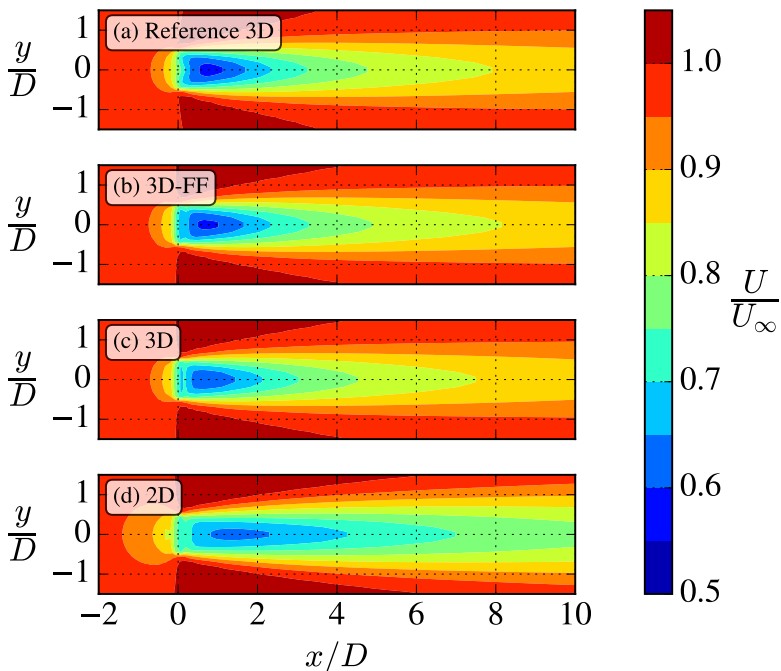

**Figure 3.** Streamwise velocity contours at hub height.

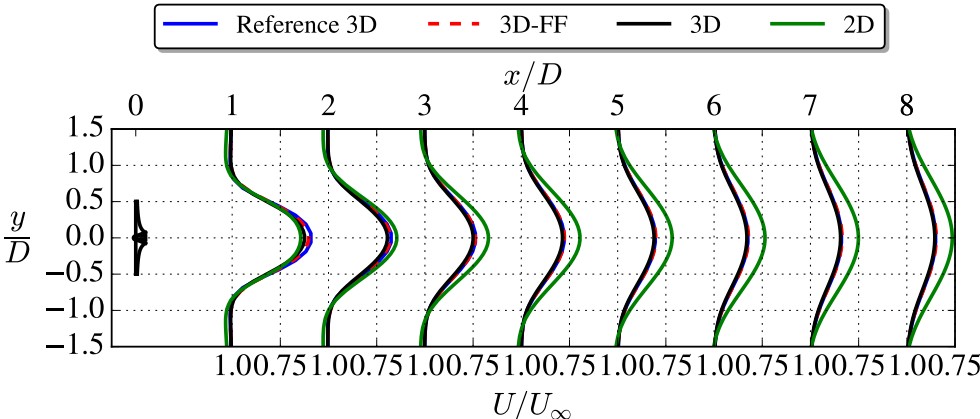

**Figure 4.** Streamwise velocity profiles at hub height.



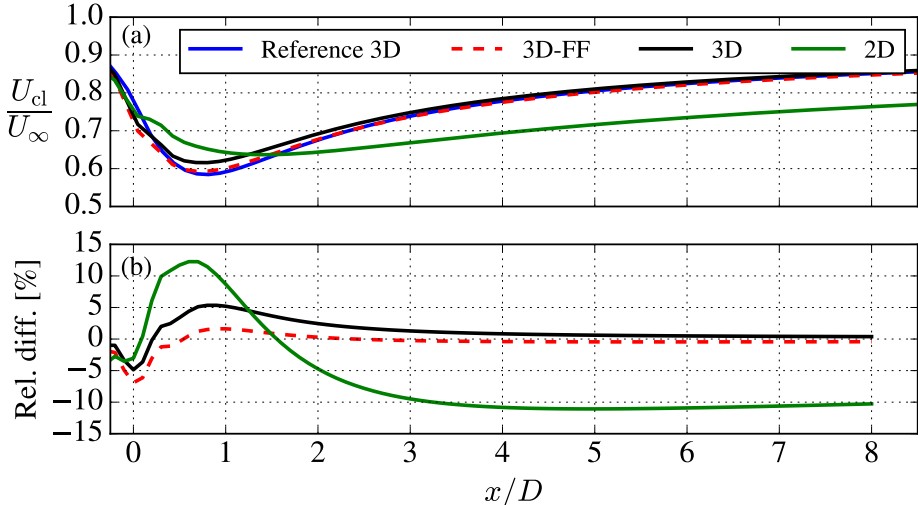

**Figure 5.** Centerline streamwise velocity through axis of AD.

For the single turbine case, the 2D OpenFOAM simulation is 717 times cheaper than the 3D OpenFOAM simulation. This
significant speed-up however comes at a cost of decreased accuracy as the turbine power is overpredicted by 3.1% compared
to 3D, while the flow fields of the 2D and 3D simulations are also substantially different. It is especially clear in Fig. 5 that
the wake in 2D recovers more slowly, which is related to the fact that high-velocity ambient flow can be entrained into the
wake from the sides only. Another visible difference is the larger induction zone in front of the turbine that originates from
the elliptic nature of the RANS equations and is amplified in 2D. This amplification is related to that the pressure disturbances
from the AD propagate further away in 2D, see Fig. 6, which is likely due to the 2D mass flow conservation (flow cannot divert
in the vertical direction). Another consequence of the change in the pressure field is a change in the balance of the pressure
gradient and shear stress divergence in Eq. 2, which directly determines the position of the minimum centerline velocity, where
$\partial U_{cl}/\partial x = 0$. In Fig. 5, it can be seen that this position changes from around $x/D \approx 0.75$ in 3D to $x/D \approx 1.5$ in 2D for the
current case, which also contributes to a delayed wake recovery.

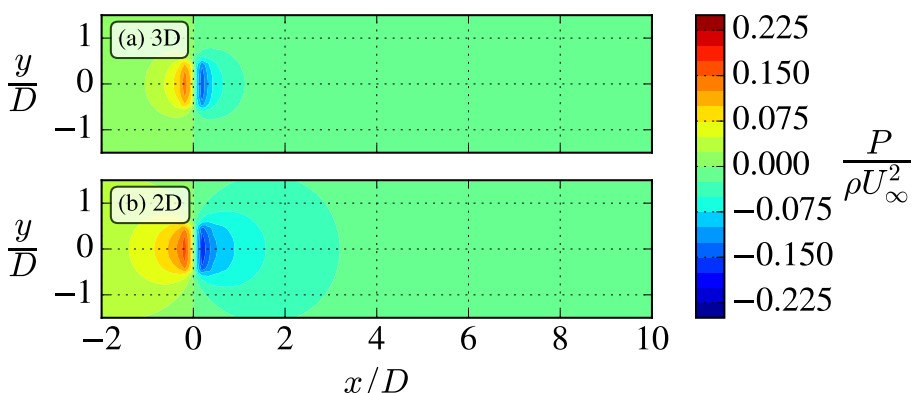

**Figure 6.** Contours of normalized pressure at hub height.

Related to the 2D mass flow conservation, although less visible, is the enhanced bypass velocity (the speed-up effect on the two sides of the turbine that makes $U/U_\infty > 1$), which can be seen in the $x/D = 1$ and $x/D = 2$ profiles of Fig. 4, and is much more clearly seen in Fig. 7. Such an enhanced bypass velocity was also reported in the 2D AD study by Boersma et al. (2018). In a related later study (van den Broek et al., 2022) it is mentioned that the width of 2D wakes are generally too wide, but as shown in Fig. 7 our simulation shows that the wake-width is only increased by a small amount in 2D. The conflicting

conclusions regarding 2D wake-width overestimation might be due to that a simple mixing length turbulence model was used in the aforementioned study.



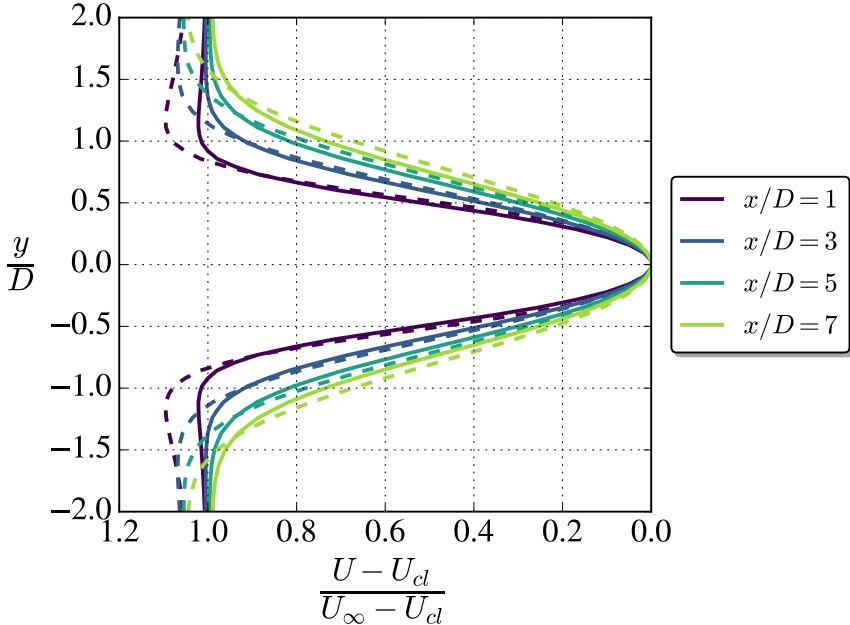

**Figure 7.** 3D (full lines) and 2D (dashed line) velocity profiles normalized by centerline velocity at each $x/D$ to highlight the shape. It is not clear from the plot, but the 2D profiles do decay to unity at large $|y/D|$.

## 4.2 Effect of turbine layout

To investigate the performance and computational cost of 2D RANS simulations in more complex scenarios, we simulate a range of different turbine layouts with an increasing number of turbines in the $x$- (along the the wind direction) and $y$-direction (perpendicular to the wind direction). The turbines are arranged in a structured layout, meaning no staggering in neither direction, see Fig. 8. In addition to $N \times N$ layouts, we also investigate layouts with more turbines in one direction than in the other, for example a 8x4 layout, hence the large number of cases in Table 4.



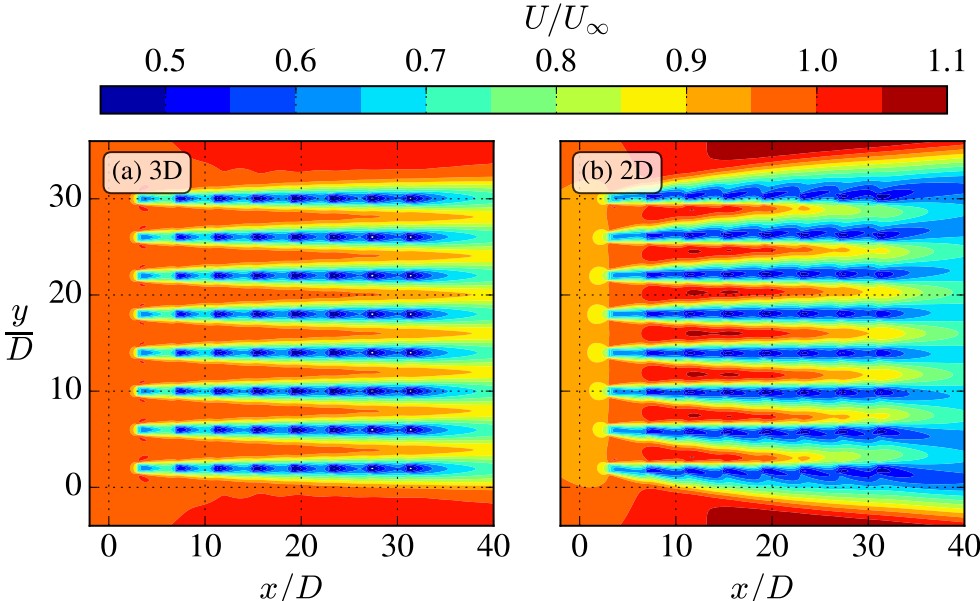

**Figure 8.** Streamwise velocity at hub height for the 8x8 turbine layout.





| Case | 3D mesh | 3D cost | 2D cost | Speed-up | $C_P^{\mathrm{3D}}$ | $C_P^{\mathrm{2D}}$ | $\frac{C_P^{\mathrm{2D}}-C_P^{\mathrm{3D}}}{C_P^{\mathrm{3D}}}$ |
|---|---|---|---|---|---|---|---|
| | [# cells] | [CPU min] | [CPU min] | [×] | [-] | [-] | [%] |
| 1x1 | 0.8M | 86 | 0.1 | 717 | 0.54 | 0.56 | 3.1 |
| 4x1 | 1.3M | 101 | 0.3 | 351 | 0.33 | 0.27 | -20.0 |
| 1x4 | 1.6M | 142 | 0.2 | 661 | 0.54 | 0.58 | 6.2 |
| 8x1 | 1.9M | 222 | 0.6 | 345 | 0.26 | 0.19 | -27.1 |
| 1x8 | 2.6M | 146 | 0.4 | 366 | 0.54 | 0.59 | 8.8 |
| 12x1 | 2.5M | 242 | 1.1 | 225 | 0.23 | 0.16 | -29.9 |
| 1x12 | 3.6M | 197 | 0.6 | 346 | 0.54 | 0.60 | 10.8 |
| 4x4 | 2.5M | 231 | 0.5 | 447 | 0.33 | 0.30 | -9.5 |
| 8x4 | 3.7M | 272 | 1.2 | 219 | 0.26 | 0.22 | -13.6 |
| 4x8 | 4.0M | 270 | 0.8 | 338 | 0.33 | 0.33 | -0.5 |
| 8x8 | 6.0M | 699 | 2.0 | 344 | 0.26 | 0.27 | 1.9 |
| 12x4 | 4.9M | 449 | 2.5 | 182 | 0.23 | 0.18 | -20.3 |
| 4x12 | 5.6M | 359 | 1.3 | 267 | 0.33 | 0.35 | 6.0 |
| 12x8 | 7.9M | 828 | 4.1 | 200 | 0.23 | 0.22 | -4.4 |
| 8x12 | 8.3M | 729 | 2.8 | 256 | 0.26 | 0.30 | 13.7 |
| 12x12 | 11.0M | 878 | 5.1 | 174 | 0.23 | 0.25 | 8.6 |

**Table 4.** Data for the low TI, $I = 5.4\%$, and dense inter-spacing, $S_x = S_y = 4D$, cases. Simulations were conducted with Intel Platinum 8628 CPUs on the ARC HPC-cluster of the University of Oxford. The farm power coefficient is defined as $C_P \equiv \frac{P_{\mathrm{farm}}}{\frac{1}{2}\rho N_x N_y \pi \left(\frac{D}{2}\right)^2 U_\infty^3}$.

From Table 4, it is clear that the computational cost of both 2D and 3D simulations, as expected, increase for increasingly large wind farms, although not monotonically with the mesh size. For example, the 1x12 simulation is cheaper than the 12x1
simulation, although the former has more computational cells. This kind of behavior is related to that in the former case the wind direction is perpendicular with the row of turbines, hence there are no direct wake-to-wake interaction, which is less computationally demanding to converge compared to the 12x1 case that has a lot of wake-to-wake interaction. Computational speed-ups, i.e. 3D cost divided by 2D cost, in the range of approximately 200 to 700 are observed for the current cases, which, although smaller than in the single turbine case, are still large savings. The speed-up is visualized in Fig. 9 along with the data
from the high TI and sparse array density cases (not included in Table 4 for brevity), which show similar speed-ups. From the plot a common trend of decreasing computational speed-up for larger $N_x$ can be observed, while it is less sensitive to $N_y$ in most cases. In other words, the computational cost of 2D simulations increases faster than the cost of 3D simulations for turbine layouts with more wake-to-wake interactions. Finally, it should be noted that there inevitably is some uncertainty in the reported computational costs, since the simulations were all run on a HPC cluster, where jobs sometimes can be allocated to
slower nodes.





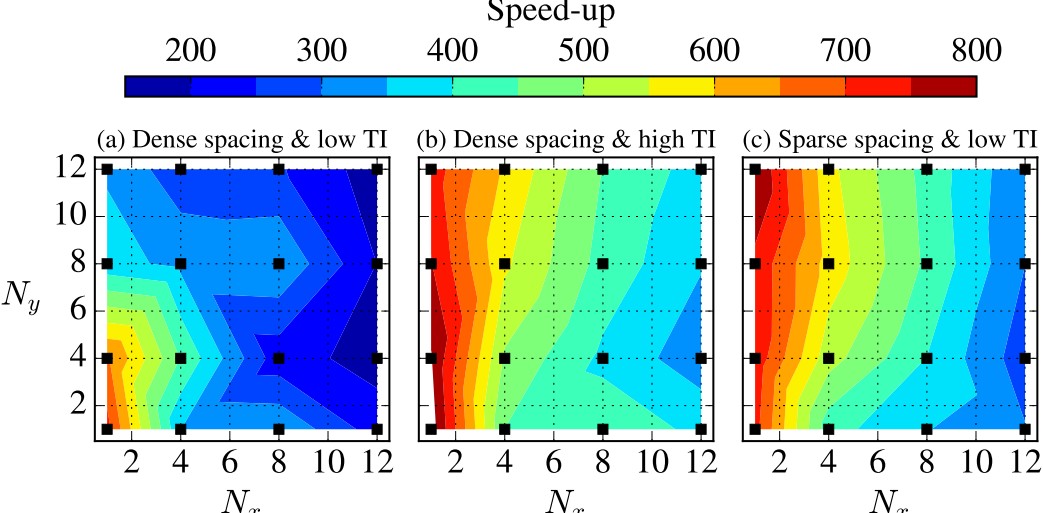

**Figure 9.** Computational speed-up (ratio of 3D cost to 2D cost) of different turbine layouts. Each square represents one wind farm simulation and the contours are plotted using these data points.

Several metrics can be used to measure the accuracy of the simulations and a fundamental metric is the total farm power, which in this study is normalized to a farm power coefficient, see definition in Table 4. In Fig. 10, it is visualized for all the different layout cases and it can be observed that the 2D simulations capture the overall trends, although with varying accuracy depending on the layout, array density, and inflow TI. In general, it can be seen that 2D simulations under-predict the farm

power for smaller farms, whereas there is a better agreement for the larger farms. The relative difference in farm power between the 2D and 3D simulations is shown as contours in Fig. 11, which more clearly show the differences. In particular, two clear trends can be identified, namely that the 2D simulations underpredict the farm power for "line"-like layouts (4x1, 8x1 and 12x1), while they overpredict farm power for "fence"-like layouts (1x4, 1x8 and 1x12). The former observation is obviously related to that the wake in 2D recovers more slowly as was found in the single turbine case. The latter is partly due to local

blockage effects (Nishino and Draper, 2015), which are negligibly small in the 3D simulations for the present turbine inter-spacings of $S_y/D = 4$ and $S_y/D = 7$, c.f. the flat 3D curves in Fig. 12c. Earlier studies indeed show that the inter-spacing needs to be less than one diameter to see any significant local blockage interaction in 3D (Baungaard, 2019) for ASL inflow, which does not have a physical boundary layer height, but is only limited by the domain height. In 2D, there is approximately a power increase of 3% for the single turbine, which is neither caused by the local blockage due to neighbouring turbines (since

there are no other turbines to interact with) nor the global blockage effect (aka. "artificial tunnel blockage", see Appendix A), and this must therefore purely be a 2D blockage effect due to 2D mass flow conservation. As more turbines are added, the local blockage effect is also present and adds to this 2D blockage effect, hence the average power is largest for the 1x12 case. As expected, the local blockage effect is less strong for larger inter-spacings $S_y/D = 7$ (sparse) compared to small inter-spacings $S_y/D = 4$ (dense).



**Figure 10.** Comparison of farm power coefficient predicted by 2D and 3D simulations.

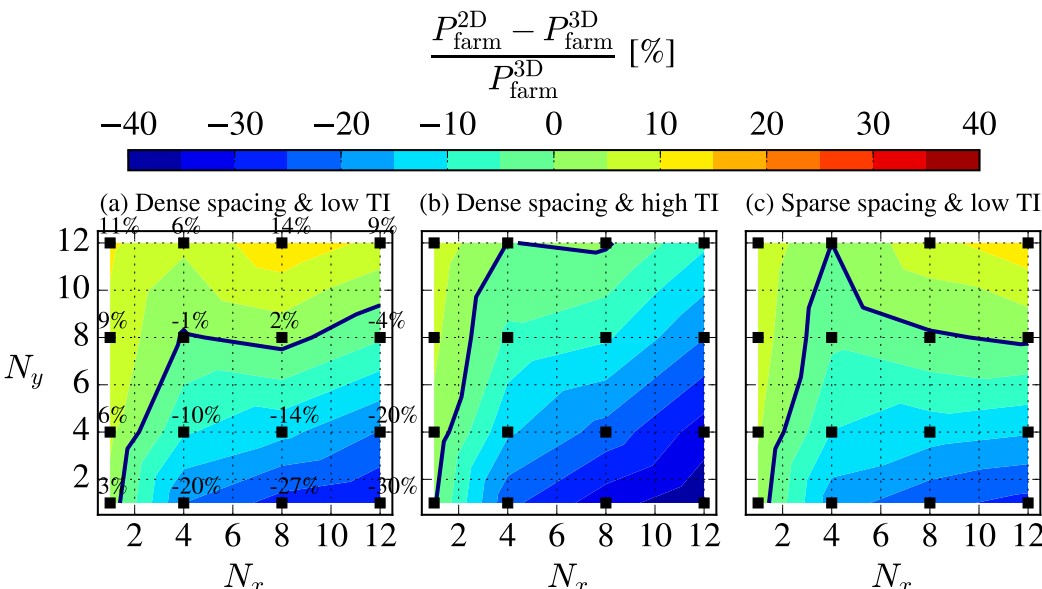

**Figure 11.** Relative difference between total farm power for 2D and 3D simulations. Full lines mark the 0-contours. Numbers from Table 4 are repeated in the left-most plot.



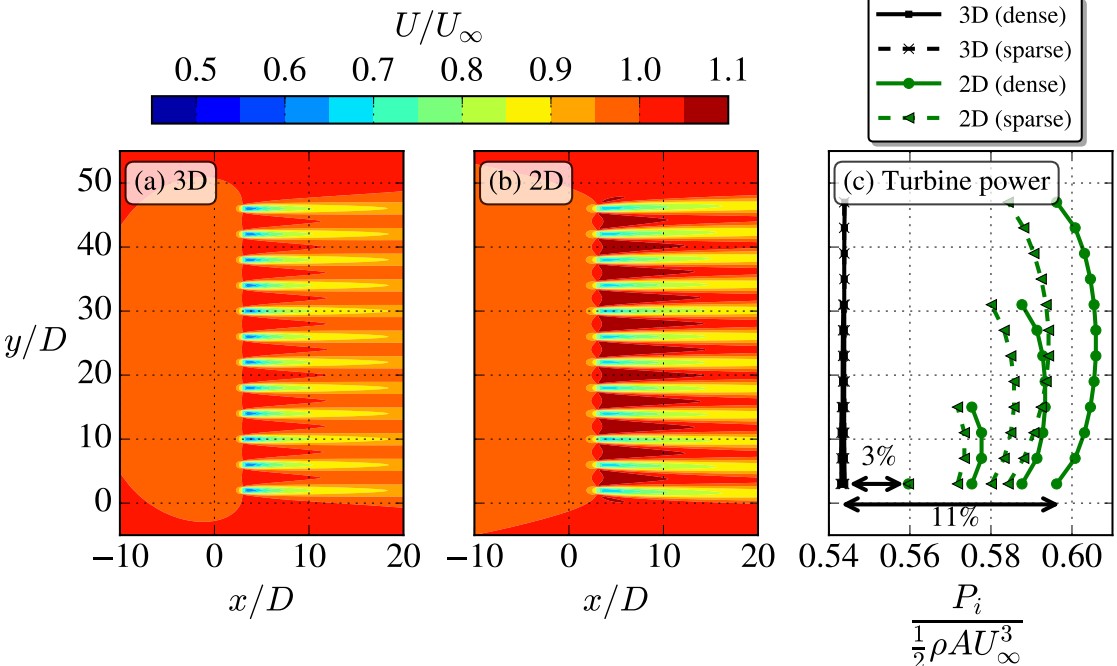

**Figure 12.** (a,b) Flow fields of the dense 1x12 farm and (c) power of individual turbines in the "fence"-like (1x1, 1x4, 1x8 and 1x12) farms, which shows the large effect of blockage in 2D.

For the intermediate cases, which are neither "line"-like nor "fence"-like, the two effects discussed in the previous paragraph will compensate each other to some extent. For example, the power is very well predicted for the 8x8 wind farm (dense spacing & low TI), where the difference in the predicted farm power between 2D and 3D simulations is only 2%. However, as revealed by Fig. 13, the power of the individual turbines are distributed significantly differently in this case for the 2D and 3D simulations, and the good agreement of the farm power, which is an integrated quantity, is therefore partially a result of error cancellations. Specifically, it can be seen that 2D model underpredicts the power of the turbines in a triangular area in the front part of the farm, whereas the turbines in the sides and back of the farm have an overpredicted power. This suggests that more effects, other than local blockage and slower 2D wake recovery, are present in this case. For example, the power in the first row of the 8x8 farm is underpredicted for the 2D simulation, which is in contrast to the 1x8 case where the power was overpredicted, and this can likely be linked to the farm blockage effect (Bleeg et al., 2018), which is known to reduce the power in the first row. This effect stems from that the incoming velocity to the farm is reduced, because the large thrust force of the farm forces some of the flow to divert around it and this type of effect is stronger in 2D, hence the underprediction. The slowdown of flow in front of the farm and the resulting large bypass velocity at the sides of the farm can clearly be seen in the flow contours of the 2D simulation in Fig. 8.

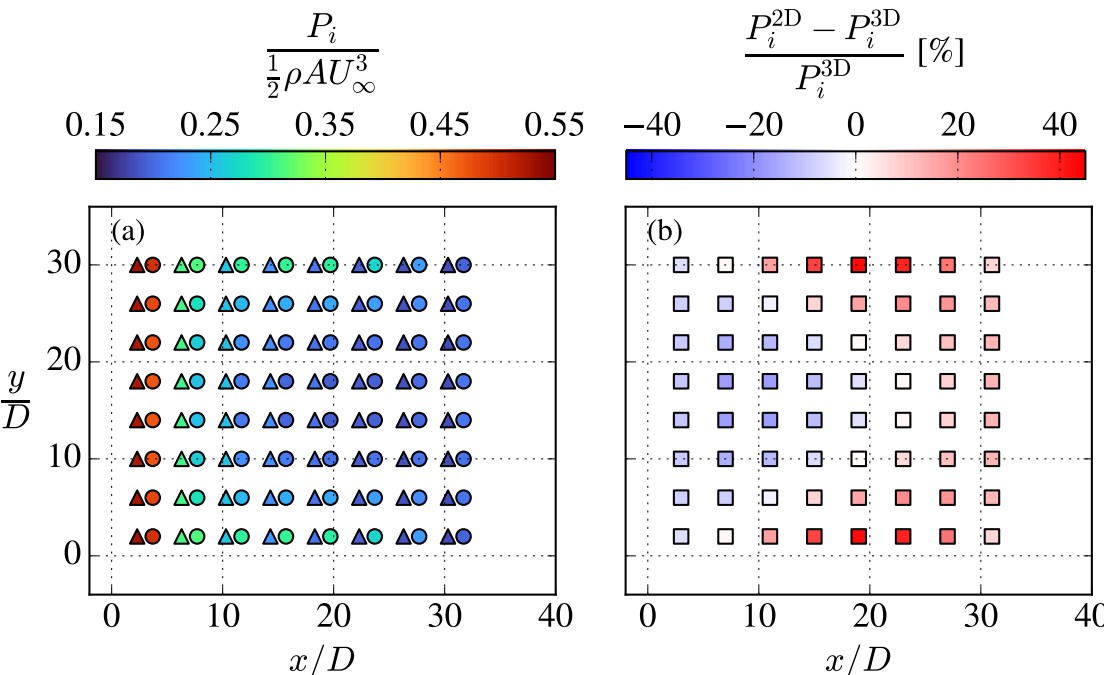

**Figure 13.** Distribution of power production in the 8x8 farm. Triangle=3D, circle=2D.

In the 8x1 case the power is consistently underpredicted for all turbines in the row summing up to a large underprediction
of farm power (27%, see Fig. 11), but this behavior is clearly not exhibited by especially the upper-most and lower-most rows
in the 8x8 farm, see Fig. 13, where the power actually is overpredicted from the third turbine and onwards. When examining
the flow contours of Fig. 8 carefully, it can be observed that the wakes in the 2D simulations are actually somewhat deflected
at the sides of farm and this must certainly be a part of the reason for the turbine power overestimations. This type of wake
deflection is clearly not present in 3D and the reason for its appearance in 2D is likely again related to the strong 2D continuity
constraint, which creates "jets" of high velocity in-between the rows. As can be seen in the flow contours, the jet in the center
of the farm is strongest and the strength then decreases as one moves towards either side of the farm, hence the flow is pushed
laterally towards each side of the farm. Regardless of the cause of the 2D wake deflection, there is no wake deflection in the
center of the farm due to mirror-symmetry of the farm, which together with the wake deflection effect at the perimeter would
explain the triangular shape of power underprediction in Fig. 13(b). Although there is no significant wake deflection for the
two middle rows, there is nevertheless a slight overprediction of power for the last two turbines in these rows, in contrast to the
8x1 case, which must therefore be caused by an effect other than 2D wake deflection and could be due to increased velocity
gradients, and thus turbulent mixing, in the wake shear layers caused by the large velocity difference between the low 2D wake
velocity and the surrounding high-velocity side jets.




### 4.3 Effect of wind direction

Until now fully-aligned cases have been considered, i.e. a wind direction of $\varphi = 0°$, where the wind farm power is heavily reduced due to turbine-wake interactions, but wind farms naturally also experience other wind directions, affecting both turbine-wake and farm-induction effects (Kirby et al., 2022). To investigate how 2D and 3D RANS simulations compare for non-aligned wind directions, a range of wind directions are run for the 8x8 and 8x4 farms, respectively. Only the wind-directions in the range $\varphi = [0, 45]°$ are simulated for the 8x8 farm (the farm has a rotational symmetry angle of 90 degrees), while the wind

directions $\varphi = [0, 90]°$ are simulated for the 8x4 farm (the farm has a rotational symmetry angle of 180 degrees). Every 5 degree in these ranges is solved. In order to leverage the rotational symmetries of the farms, it is also a requirement that the AD model does not employ tangential forces and that the inflow is non-veered (van der Laan et al., 2022), which are both satisfied in our setup.

To simulate the different wind directions, the farm layout is rotated rather than changing the actual flow direction, hence the

flow will be aligned with the $x$-direction for all wind directions, see example in Fig. 14. This is a convenient technique for several reasons, (i) the same grid can be used for all wind directions, (ii) the main flow is parallel with the grid lines, which is numerically advantageous, and (iii) the simulations can be run consecutively to save computational time (van der Laan et al., 2022), which however is not utilized in the current study in order to be able to compare the computational cost of each wind direction. An inner domain is chosen such that it can contain all the turbines for the wind direction with the widest layout,

which means that the mesh for the 8x8 simulations is larger (9.4M cells) than the grid which was used for the 8x8 farm in the previous section (6.0M cells, see Table 4). The same applies to the 8x4 simulations.

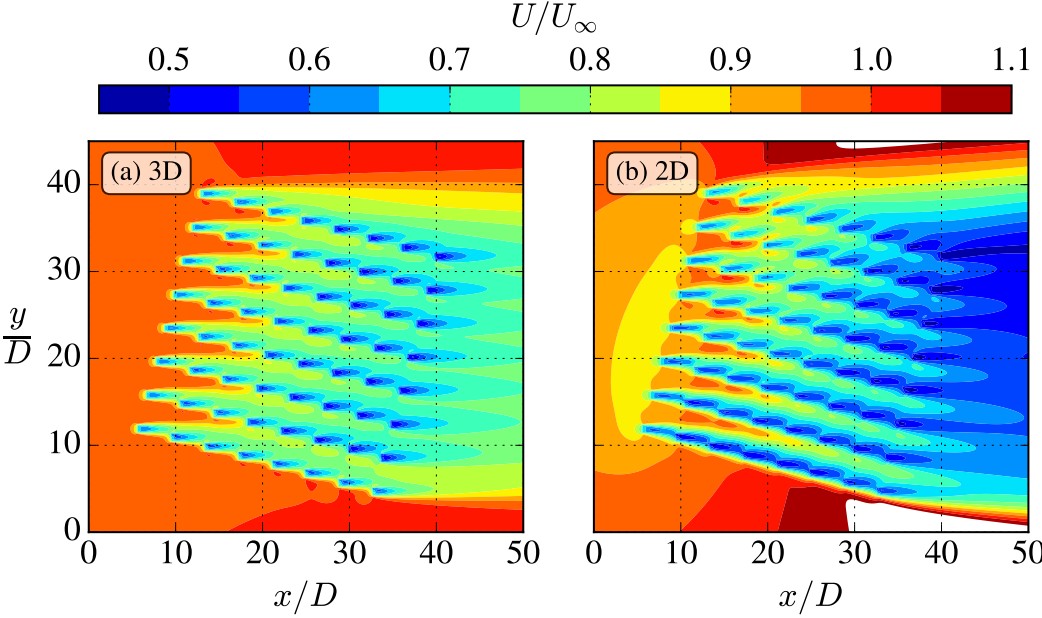

**Figure 14.** Streamwise velocity contour for 8x8 farm at wind direction $\varphi = 15°$.



The computational speed-up of the 2D compared to 3D simulations for the varying wind directions is on the order of 300-500 for the two layouts considered, see Table 5, which is similar to the aligned cases in the previous section. The costs of the 3D simulations are fairly independent of the wind direction, while the costs of the 2D simulations are more dependent. Especially for the 8x4 farm, there is a clear trend of decreasing cost with increasing wind direction up to $\varphi = 90°$, which again highlights the observation from Fig. 9 that there is a larger speed-up for "fence"-like layouts compared to "line"-like layouts. As stated earlier, the simulations were run on a HPC cluster, which adds some uncertainty to computational times, hence the speed-ups reported in Table 5 are only meant to give a sense of the computational cost differences rather than exact numbers.

| $8 \times 8$ farm | | | | $8 \times 4$ farm | | | |
|---|---|---|---|---|---|---|---|
| Case | 3D cost | 2D cost | Speed-up | Case | 3D cost | 2D cost | Speed-up |
| [deg] | [CPU min] | [CPU min] | [×] | [deg] | [CPU min] | [CPU min] | [×] |
| 0 | 1553 | 4.3 | 362 | 0 | 965 | 2.9 | 335 |
| 5 | 1553 | 4.8 | 322 | 10 | 955 | 2.7 | 357 |
| 10 | 1434 | 4.2 | 344 | 20 | 951 | 2.6 | 364 |
| 15 | 1429 | 3.7 | 386 | 30 | 1017 | 2.6 | 389 |
| 20 | 1434 | 3.9 | 366 | 40 | 952 | 2.5 | 379 |
| 25 | 1455 | 4.3 | 339 | 50 | 954 | 2.3 | 409 |
| 30 | 1434 | 4.0 | 354 | 60 | 960 | 2.2 | 434 |
| 35 | 1501 | 4.3 | 350 | 70 | 937 | 2.0 | 471 |
| 40 | 1404 | 4.3 | 324 | 80 | 967 | 2.1 | 470 |
| 45 | 1500 | 4.7 | 318 | 90 | 999 | 2.0 | 495 |

**Table 5.** Computational cost for varying wind direction. Only every second entry of the 8x4 case is shown.

The normalized farm power coefficient is displayed for the range of studied wind directions in Fig. 15, which shows that the main trends can be predicted with the 2D simulations. The minimum power is obviously at $\varphi = 0°$ and it is also clear that there should be a power dip at $\varphi = 45°$, when the wind direction is "diagonally" aligned with the turbines in the farm. There are however more wind directions at which the turbines are aligned in a structured farm layout, namely $\varphi = \arctan(n_y/n_x)$, where $1 < n_x < N_x - 1$ and $1 < n_y < N_y - 1$. This explains the small dip in power at $\varphi = 25°$, since $\varphi = \arctan(1/2) \approx 26.6°$, which is visible for the 3D simulations, but not captured by the 2D simulations. In the previous section it was shown that there was an excellent agreement of only 2% difference between the 2D and 3D simulations for the 8x8 farm at $\varphi = 0°$, but from Fig. 15 it is clear that this is generally not the case for $\varphi \neq 0°$. For example, there is an underprediction of 17% at $\varphi = 15°$ and in Fig. 16 it can indeed be seen that the individual turbine power is underpredicted in the majority of the farm, which is notably different from the $\varphi = 0°$ case, see Fig. 13, where there was an overprediction at the two sides of the farm. This change can be linked to the change in wind farm flow of the $\varphi = 15°$ case, which, as shown in Fig. 14, does not have high-velocity jets between the rows as in the $\varphi = 0°$ case. Furthermore, the flow is not symmetric around the middle rows anymore and the center of the induction zone is shifted downwards, which explains why the 2D wake deflection is more pronounced in the upper part





of the farm flow, c.f. Fig. 14b. In general, however, it cannot be concluded that there is a worse agreement for non-aligned wind directions than for aligned wind directions, as exemplified by the 8x4 case in Fig. 14, where there is an excellent agreement for the wind direction range $45° < \varphi < 90°$. This again highlights the importance of the turbine layout, since the wind directions

$\varphi > 45°$ represent more "fence"-like layouts, while the wind directions $\varphi < 45°$ represent more "line"-like layouts, and, as discussed in the previous section, the 2D flow physics and farm power predictions are significantly different for these two scenarios.

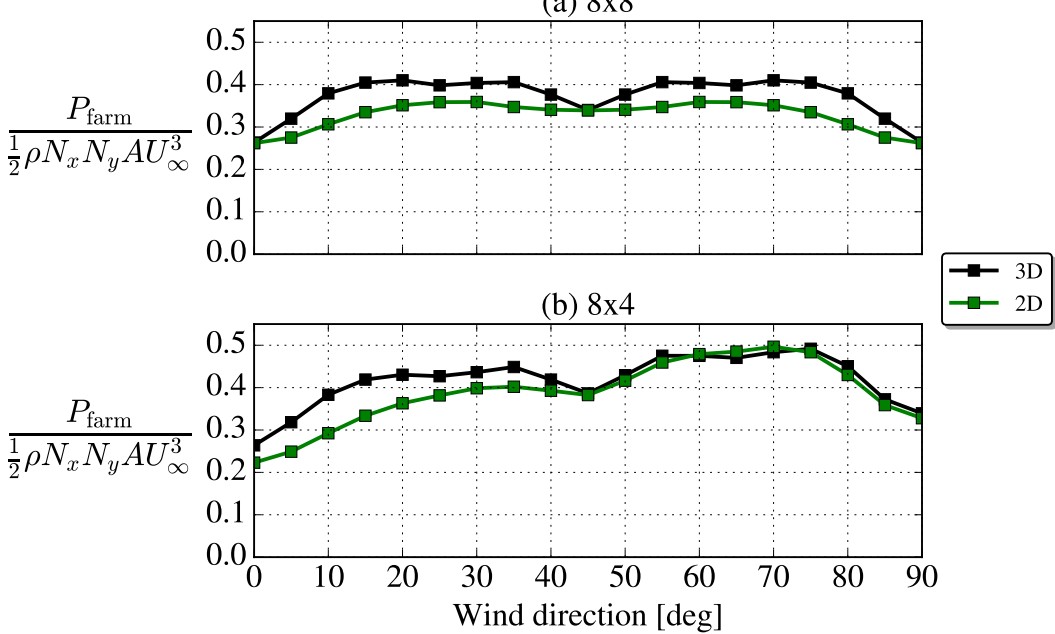

**Figure 15.** Farm power coefficient. The values for the 8x8 farm at $\varphi = [50, 90]°$ have been mirrored from the $\varphi = [0, 40]°$ values.



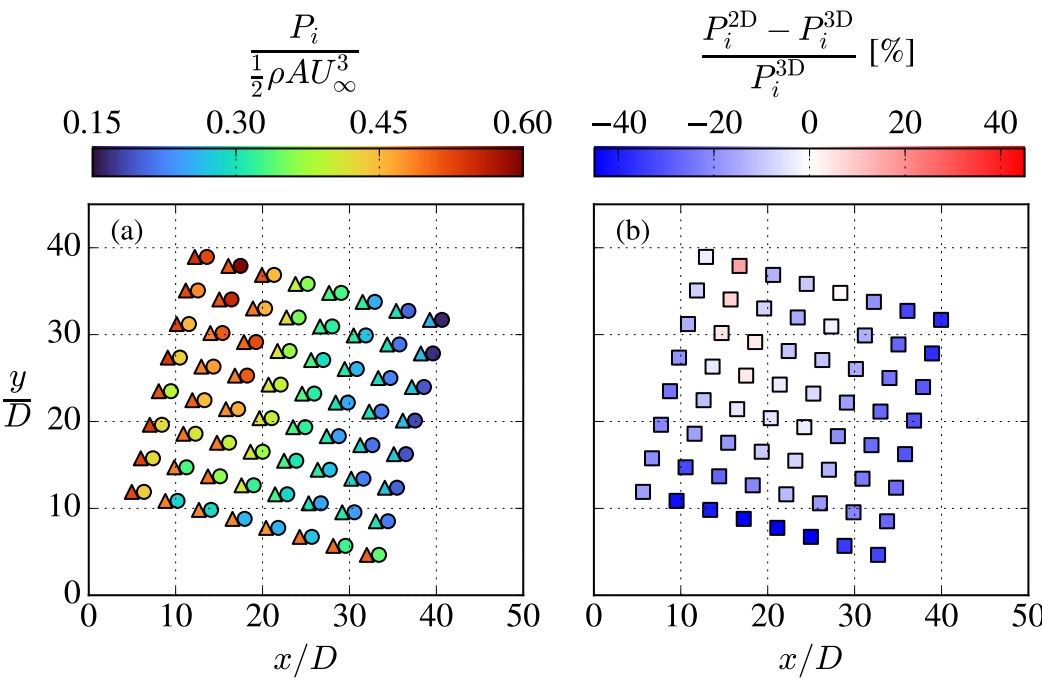

**Figure 16.** Power distribution in the 8x8 farm with wind direction $\varphi = 15°$. The farm power is underpredicted by 17% in this case.

## 5 Conclusions

A systematic study on the use of 2D RANS simulations for wind farm flows has been conducted. The horizontal plane at hub-
height was chosen for the 2D simulations as this allowed the study of different turbine layouts. In contrast to 3D simulations,
it is necessary to include additional source terms in the 2D turbulence equations to avoid unphysical decay of turbulence. The
force and power of the wind turbines, modeled as actuator disks in this work, can with minor modifications also be applied in
2D simulations. No other corrections were added and the study thus constituted a simple direct comparison between 2D and
3D RANS wind farm flow simulations.

The computational cost and accuracy compared to full 3D RANS simulations have been reported for a wide range of turbine
layouts and wind directions, and variations thereof with larger inter-spacing and higher turbulence intensity, respectively. For
these specific cases, it is found that the 3D simulations require on the order of 200 to 700 times more CPU time than their
corresponding 2D simulations. This speed-up obviously depends on several parameters, but since the 2D and 3D simulations
are run with the same code, numerical setup, and a relatively low number of vertical grid points ($N_z = 46$), it appears safe to
conclude that 2D simulations, in general, are at least two orders of magnitude faster than 3D simulations for wind farm flows.

As a first test of accuracy, a single turbine is studied, where it is found that the 2D wake recovers more slowly, but has a
similar wake width as the corresponding 3D case. The flow is more constricted in 2D, which leads to a larger induction zone, a
higher bypass-velocity (the flow acceleration at the sides of the turbine), and a slight increase of turbine power of approximately





3%. For the more complicated wind farm cases with multiple turbines, it is found that the 2D simulation captures the correct
trend of farm power, but that the difference compared to the corresponding 3D simulation can be anywhere in the range of
$-30\%$ to $15\%$. The largest underprediction is seen for "line"-like layouts, i.e. a farm with only one row of turbines that is
aligned with the wind direction, where the power is under-predicted due to the slow 2D wake recovery. On the other hand,
the farm power is overpredicted for "fence"-like layouts, i.e. a farm with one row of turbines that is perpendicular with the
wind direction, due to enhanced blockage effects in 2D. In between these two extreme cases, the agreement is generally better,
which however partly can be attributed to error cancellations between the two aforementioned effects, as well as the farm-scale
blockage effect, a wake deflection effect in the perimeter of the farms, and the increased wake shear layer velocity gradients.
The complex balance of all these effects change when the wind direction is not aligned with the rows of turbines, which can
lead to worse or better agreements with the 3D simulations depending on the layout. Despite these challenges, it is encouraging
that the overall flow and turbine power trends can be replicated with the much cheaper 2D simulations.

Since there is a clear correlation between the turbine layout and the under- and over-prediction of farm power, one could
possibly construct an empirical model to correct this. However, we see it as a more promising path to instead modify the
governing equations to account for the various 2D effects and thus improve the wind farm flow prediction and thereby also
the farm power prediction. This would possibly also enable 2D simulations to be used for studying farm wakes, which the
model is not recommended for in its current state due to its slow wake recovery. Modifications would need to be made to both
the continuity, momentum and turbulence equations to address all the relevant 3D-to-2D effects. A simple correction for the
continuity equation has already been suggested by Boersma et al. (2018), but since it is based on an axisymmetry assumption,
its general validity is not clear and should be tested more thoroughly.

In conclusion, large computational speed-ups and sensible farm flow and power predictions can be achieved with 2D RANS
simulations. It is however recommended to include further modifications to the governing equations to achieve better agreement
with the full 3D simulations. The 2D simulation setup presented in the current work should therefore mainly be seen as a
baseline or proof-of-concept setup.

*Code and data availability.* A repository with the OpenFOAM implementation of the AD used in this work is available at https://github.com/
mchba/actuatorDiskFoam. The 2D RANS simulations in this article were run with the OpenFOAM v2206 CFD code through the Python
framework, `farm2d`, which is freely available at https://github.com/mchba/farm2d, and can be used to reproduce the data presented in this
study. The 3D simulations were run through a similar Python framework and a selection of the data (the 1x1, 1x12 and 8x8 farms) is available
at https://doi.org/10.5281/zenodo.15046923 (Baungaard, 2025).

## Appendix A: Grid sensitivity study

The 3D single turbine case is simulated with increasingly fine resolution, $D/\Delta x = D/\Delta y = \{4, 8, 16, 32\}$, in the wake region,
see Fig. 1, and the stretched vertical grid is likewise refined. The computational cost of the simulations are $[0.1, 1.4, 19.4, 225.3]$
CPU hours, respectively. As is apparent from Fig. A1-A2, the velocity field of the coarse $D/4$ simulation significantly deviates



from the finer resolved simulations in the near-wake, but has a good agreement in the far-wake. This has been further quantified with a Richardson extrapolation (RE) (Réthoré et al., 2014) of the centerline velocity, which shows that the discretization error drops to below 1% (c.f. Fig. A1b) at $x/D = 3$ for the $\Delta = D/4$ simulation. The $D/8$ simulation still has a slightly different near-wake velocity profile from the finer resolved simulations, but is nevertheless chosen for the current study due to its

good compromise between accuracy and computational cost. This is also a typical grid resolution used for RANS wind farm simulations. The same resolution is used for the 2D simulations in order to be able to make direct comparisons with their 3D counterparts.

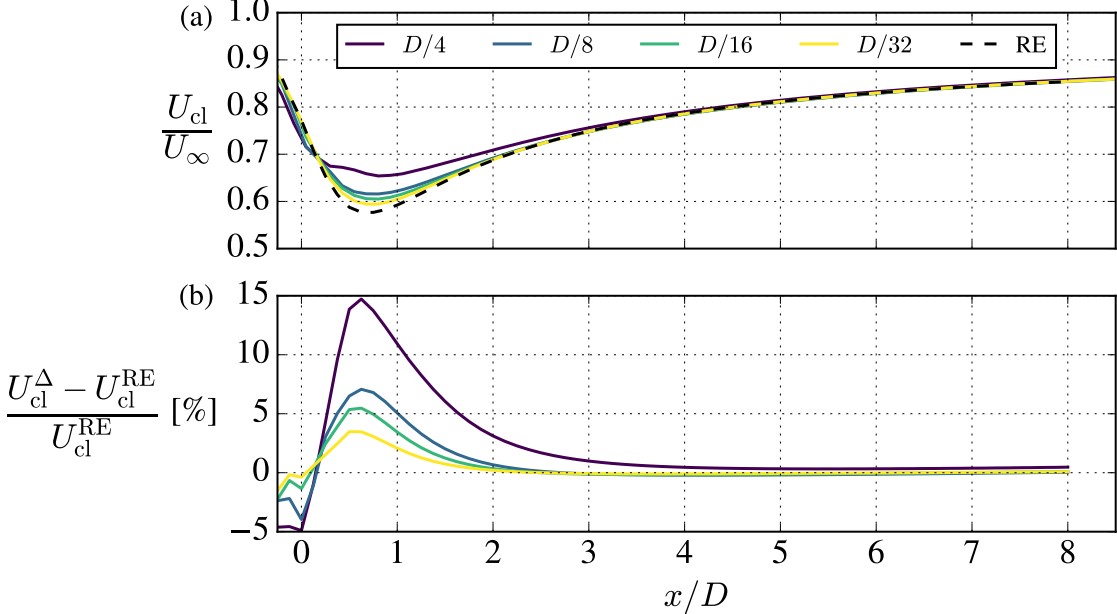

**Figure A1.** Streamwise velocity at the centerline through the AD.



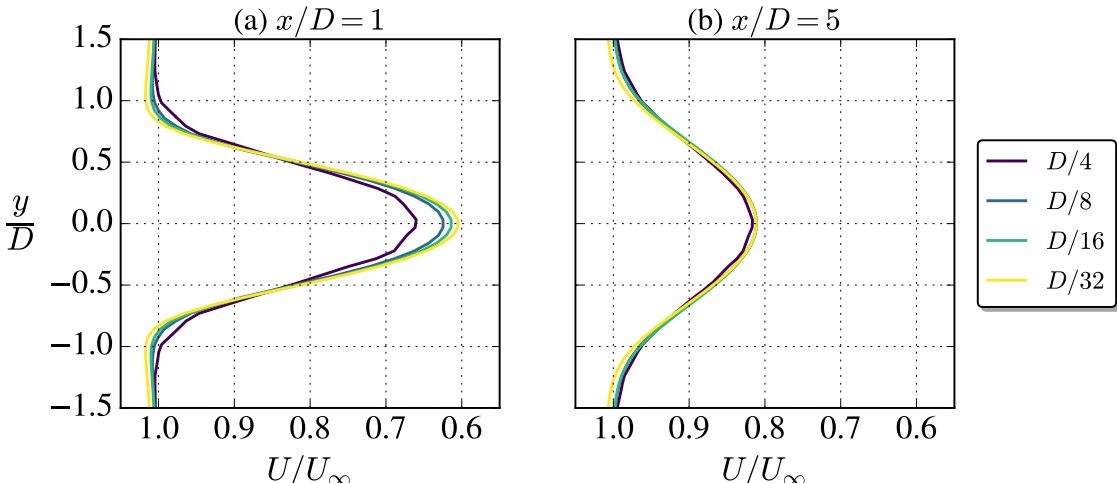

**Figure A2.** Profiles of streamwise velocity at hub height at two downstream positions.

An AD model based on the disk-averaged velocity is used in this work, see Sect. 2.4, and the power and thrust of the AD is therefore not fixed a-priori, but is coupled with the flow solution. The full iteration history of the power and thrust coefficients

are given in Fig. A3 for each simulation, which show that the simulations indeed have converged to steady-state solutions at the end of the iterations. A residual criterion of $10^{-3}$ is used for all governing equations, which can thereby be confirmed to be sufficiently strict to converge the simulations. It is well-known (van der Laan et al., 2015b) that ADs based on disk-averaged velocity overpredict the 1D momentum theory values of the thrust and power coefficients, but this is actually only observed for the two finer resolved simulations in Fig. A3. This is likely related to that the disk in our current OpenFOAM

setup cannot be refined independently of the background mesh, as is for example possible in the EllipSys3D code (Sørensen, 1995; Réthoré et al., 2014; Baungaard, 2019). Aside from this deficiency, it is clear that the code still predicts values close to the 1D momentum theory, which verify the current 3D AD implementation.



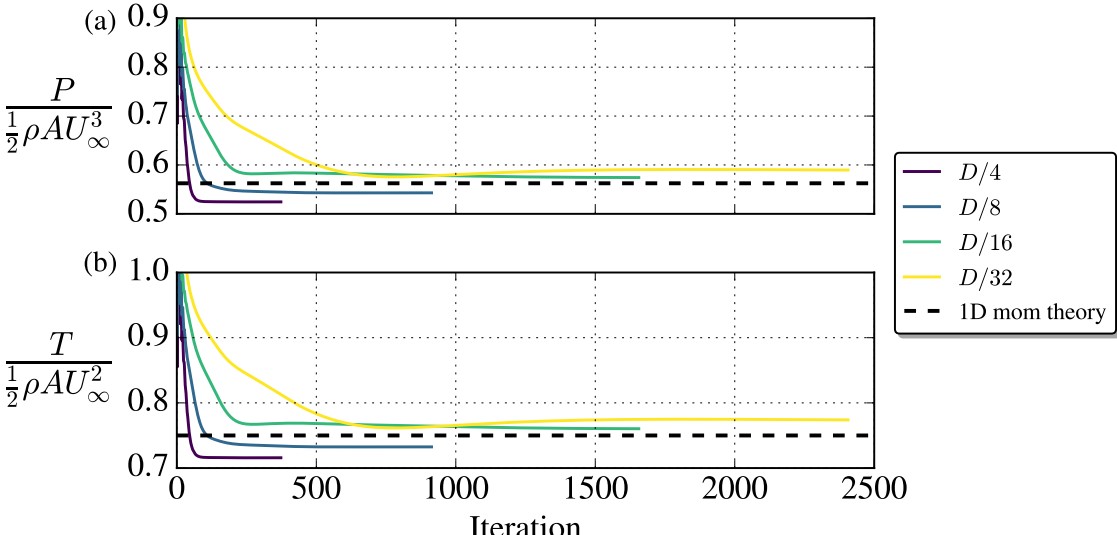

**Figure A3.** Iteration history of normalized turbine power and thrust.

Blockage effects are amplified in 2D and a parametric study of the domain width has therefore been conducted for the $D/8$ simulation to assess how wide the domain needs to be to avoid the artificial tunnel blockage effect. Domain widths of $L_y/D = \{14, 24, 54, 104, 204\}$ are simulated corresponding to an outer domain buffer of $L_{y,n}/D = L_{y,s}/D = \{5, 10, 25, 50, 100\}$, respectively, with the width of the inner domain being fixed to $l_y/D = 4$, see Fig. 1. The power of the turbine is used to measure the sensitivity of domain width, see Fig. A4. There is a relatively weak dependence on the domain width in the 3D case, whereas a much stronger sensitivity is seen in 2D with a difference of 2.2% between the narrowest and widest domains. It is chosen to use an outer domain buffer of $50D$ (corresponding to $L_y/D = 104$ for the single turbine case) for all cases in the current work, for both the 2D and 3D simulations, in order to avoid the artificial tunnel blockage effect.





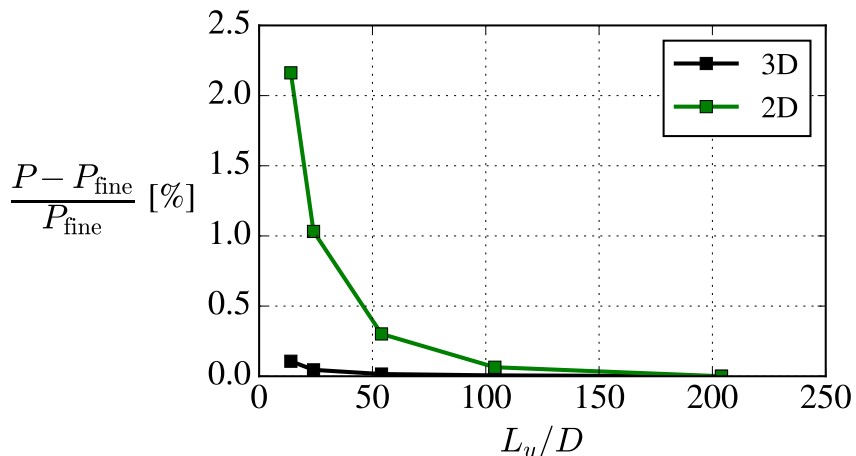

**Figure A4.** Sensitivity of AD power to domain width. The setup is the same as in the $D/8$ simulation, except that the width of the domain is varied.

## Appendix B: Verification of 2D AD implementation

By setting the stress divergence term to zero in the RANS momentum equation, Eq. 2, one obtains the Euler equations

$$\frac{\partial U_j}{\partial x_j} = 0, \tag{B1}$$

$$U_j \frac{\partial U_i}{\partial x_j} = -\frac{1}{\rho}\frac{\partial P}{\partial x} + f_i. \tag{B2}$$

From these equations, an analytical solution was derived for 2D AD flow by Madsen (1988), which is valid for $C_T \ll 1$. The normalized pressure and streamwise velocity of the solution are

$$\frac{P(x,y)}{\rho U_\infty^2} = -\frac{C_T}{4\pi}\left(\tan^{-1}\left(\frac{D/2 - y}{x}\right) + \tan^{-1}\left(\frac{D/2 + y}{x}\right)\right), \tag{B3}$$

$$\frac{U(x,y)}{U_\infty} = \begin{cases} 1 - \frac{P}{\rho U_\infty^2}, & \forall x, y \notin S_{\text{wake}} \\ 1 - \frac{P}{\rho U_\infty^2} - \frac{1}{2}C_T, & \forall x, y \in S_{\text{wake}} \end{cases} \tag{B4}$$

where $S_{\text{wake}} = \{x > 0 \wedge y < D/2 \wedge y > -D/2\}$ is the wake region. This solution is useful for code verification and has been

used for this purpose in previous studies by Erek (2011) and Réthoré and Sørensen (2012). A slight disagreement with the solution was found by Erek (2011), which however is most likely caused by a too small domain.

A case with $C_T = 0.01$ and numerical setup as described in Sect. 2.2, except with a fine grid resolution, $D/\Delta x = D/\Delta y = 32$, is simulated. An excellent agreement with the analytical solution is found, see Fig. B1, except a small wiggle in the velocity at the AD position. This is a well-known problem of applying discrete body forces in collocated finite-volume codes (analogous



to the pressure-velocity coupling problem), and could be alleviated by applying a more sophisticated force allocation method (Réthoré and Sørensen, 2012), but is not pursued in the current work. Apart from this small problem, we conclude that the 2D AD model in OpenFOAM works as expected.

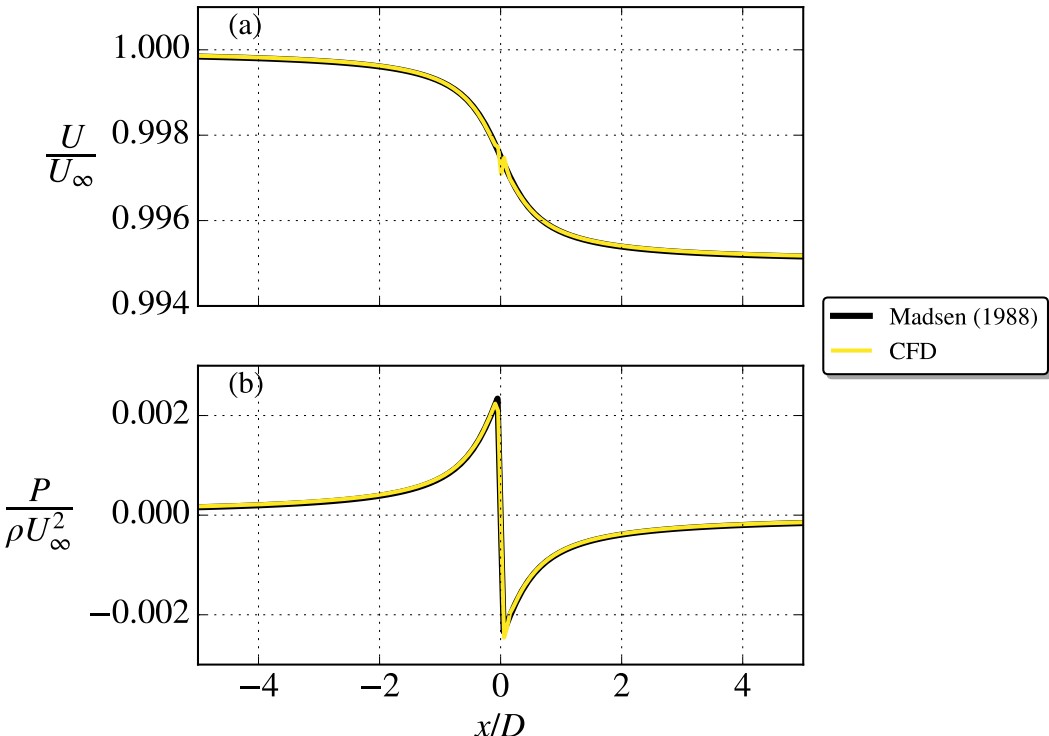

**Figure B1.** Centerline velocity (top) and pressure (bottom). Reference 2D AD solution from Madsen (1988) with $C_T = 0.01$.

*Author contributions.* All authors contributed equally to the initial idea and discussion of 2D RANS wind farm simulations. MB ran the simulations, post-processed the results, and wrote the first draft of the paper. The article was internally reviewed by TN and MPVDL, and
finalized by all authors.

*Competing interests.* The authors declare that they have no conflict of interest.

*Acknowledgements.* This work has been supported by the Carlsberg Foundation (grant CF23-1002). We also acknowledge the computational resources granted by the ARC HPC service at the University of Oxford.





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
