# Peer review of "Simulating wind farm flows at hub height with 2D Reynolds-averaged Navier-Stokes simulations"

_Wind Energy Science, 2025_

## Referee Comment (RC1)

**Review of the manuscript wes-2025-50, "Simulating wind farm flows at hub height with 2D Reynolds-averaged Navier-Stokes simulations", by M. Baungaard, T. Nishino, M. P. van der Laan.**

This manuscript explores the use of 2D RANS simulations to predict wind farm power. Computational costs and accuracy of 2D RANS simulations are compared against 3D RANS simulations, confirming a reduction in computational cost of almost 2 orders of magnitude. However, significant errors in power capture and wake evolution are identified. Therefore, further modifications of the governing equations are recommended for future work.

This reviewer believes that the contribution of the manuscript is rather limited, and it only consists in adding two source terms in the $k$-$\varepsilon$ equations (Eqs. 11 and 12) to avoid unrealistic turbulence decay in the streamwise direction. However, these terms have already been used in previous works, e.g., van der Laan *et al.* 2015, 2017, for a similar purpose. I think the main modeling challenges in performing 2D RANS simulations of wind farms are only quickly mentioned in the manuscript and never tackled directly; instead, they are mentioned as future work. For instance, for the calculation of the equivalent 2D thrust force from a 3D actuator disk, the approach proposed by Madsen 1988 is used, even though the limitations associated with the assumption of a 3D axisymmetric actuator disk are recalled. As mentioned by the authors, major issues are associated with the evolution of wakes and the presence of unrealistic speedups due to the lack of vertical mass fluxes in the continuity equation, and of momentum fluxes, turbulent fluxes, and dispersive stresses in the momentum equation. Modeling these contributions as a function of the background boundary layer and wind farm characteristics is one of the main challenges for 2D RANS simulations of wind farms. In summary, I would recommend the authors to keep working in this direction and tackling these important research topics.

**Comments:**
1.      L8 – "… while the predicted farm power is within −30% to 15% for all cases…", I guess you mean the average percentage error. Is this error calculated against experimental data, LES, 3D RANS? Is it at the turbine level or farm level (I later understood from Fig. 11 and related text that it is at the farm level)? Please be more specific.
2.      Fig. 3 and more in general for all the figures– this caption should be improved by describing each panel.